# *Histoplasma capsulatum* Activates Hematopoietic Stem Cells and Their Progenitors through a Mechanism Dependent on TLR2, TLR4, and Dectin-1

**DOI:** 10.3390/jof8101108

**Published:** 2022-10-20

**Authors:** Carolina Rodríguez-Echeverri, Beatriz L. Gómez, Ángel González

**Affiliations:** 1Basic and Applied Microbiology Group (MICROBA), School of Microbiology, Universidad de Antioquia, Medellín 050010, Colombia; 2Translational Microbiology and Emerging Diseases Research Group (MICROS), School of Medicine and Health Sciences, Universidad del Rosario, Bogotá 111221, Colombia

**Keywords:** hematopoietic stem cells, HSC, hematopoietic stem cells and progenitors, HSPC, *Histoplasma capsulatum*, immune response

## Abstract

Hematopoietic stem cells (HSCs), a multipotent and self-renewing population responsible for the generation and maintenance of blood cells, have been the subject of numerous investigations due to their therapeutic potential. It has been shown that these cells are able to interact with pathogens through the TLRs that they express on their surface, affecting the hematopoiesis process. However, the interaction between hematopoietic stem and progenitor cells (HSPC) with fungal pathogens such as *Histoplasma capsulatum* has not been studied. Therefore, the objective of the present study was to determine if the interaction of HSPCs with *H. capsulatum* yeasts affects the hematopoiesis, activation, or proliferation of these cells. The results indicate that HSPCs are able to adhere to and internalize *H. capsulatum* yeasts through a mechanism dependent on TLR2, TLR4, and Dectin-1; however, this process does not affect the survival of the fungus, and, on the contrary, such interaction induces a significant increase in the expression of IL-1β, IL-6, IL-10, IL-17, TNF-α, and TGF-β, as well as the immune mediators Arg-1 and iNOS. Moreover, *H. capsulatum* induces apoptosis and alters HSPC proliferation. These findings suggest that *H. capsulatum* directly modulates the immune response exerted by HPSC through PRRs, and this interaction could directly affect the process of hematopoiesis, a fact that could explain clinical manifestations such as anemia and pancytopenia in patients with severe histoplasmosis, especially in those with fungal spread to the bone marrow.

## 1. Introduction

Hematopoietic stem cells (HSC), defined by their ability to differentiate to multiple lineages and their extensive self-renewal capacity, are responsible for the hematopoiesis process. This process involves the production of all mature blood cells, which in turn contributes to the maintenance of homeostasis and cellular restoration after infectious processes or therapeutic ablation [1,2]. The population of HSCs has been fully characterized, and these cells are known as LKS (Lin^−^, c-Kit^+^, and Sca-1^+^) cells, based on the differential expression of the tyrosine kinase c-Kit receptor (CD117) and the Sca-1 glycoprotein membrane and the lack of specific lineage markers (Ter119, Gr-1, Mac-1, B220, CD4, and CD8) [3].

In recent years, hematopoietic stem and progenitor cells (HSPC) have gained great importance in the study of infections caused by pathogens, due to their differentiation properties and proliferation capacity [4]. In addition, HSPCs have been described as expressing functional pattern recognition receptors (PRRs) on their surface, such as *Toll*-like receptors (TLR), including mainly TLR2 and TLR4, and C-like lectin receptors such as Dectin-1, among others [5,6], which are directly involved in the recognition of pathogens and the activation of the signaling pathways involved in the induction of proliferation and differentiation to macrophages during infection [3]. Similarly, HSPCs can recognize soluble mediators secreted by cells of the immune system or stromal cells activated by pathogens, which stimulate increased migration to the infected tissues [3,7]. This direct interaction of the bone marrow cells with the invading pathogen or activation via the production of cytokines in response to the infectious challenge is a process known as “emergency hematopoiesis” and occurs mainly during systemic infections [8].

In this sense, histoplasmosis is a systemic and endemic mycosis of great importance in the world. The infection is acquired by inhaling infecting particles that include microconidia and small fragments of the thermally dimorphic fungal species of the genus *Histoplasma*, among which the phylogenetic species *H. capsulatum* sensu stricto, *H. mississippiense*, *H. ohiense*, *H. suramericanum*, and *H. africanum* have been described [9]. The infection can be asymptomatic or can develop into a serious and fatal disease mainly affecting the lungs [10]; the clinical presentation depends, in turn, on the amount of the inhaled inoculum and the immunological status of the patient. Patients with immunological alterations can develop dissemination to the bone marrow with the development of anemia and pancytopenia, which indicates a significant compromise of the progenitor cells present in the bone marrow niche [11]. Currently, despite the fact that the mechanisms of expansion and differentiation of HSPC have been extensively studied, the mechanisms by which these cells respond to invading fungal pathogens are poorly understood. On these lines, some studies have reported that HSPCs have the ability to interact with fungal pathogens; thus, *Candida albicans* is capable of activating HSPCs and inducing their proliferation and differentiation to myeloid lineage specifically to macrophages, through interaction with TLR2 [12,13].

In the present study, the interaction between HSPCs and *H. capsulatum* yeasts was investigated using in-vitro assays. It was established that these cells have the capacity to phagocytize the fungal yeasts, although this process did not affect the survival of the fungus. The interaction of *H. capsulatum* with HSPCs also induces a significant increase in the expression of TLR2 and Dectin-1 but not of TLR4. Additionally, this interaction increases the expression of IL-1β, IL-6, IL-10, IL-17, TNF-α, and TGF-β, as well as the immune mediators Arg-1 and iNOS. It was also established that the fungal yeasts induced apoptosis and altered the proliferation of HSPCs.

## 2. Materials and Methods

### 2.1. Ethical Considerations

This study was carried out in accordance with the regulations set forth in Colombian Law 84/1989 and Resolution No. 8430/1993, as well as those issued by the European Union and the Canadian Council for Animal Care. The protocol was approved by the Ethics Committee for Experimentation with Animals of the Universidad de Antioquia (Act No. 120).

### 2.2. Isolation, Purification, and Maintenance of Bone Marrow-Derived HSPCs

HSPCs were obtained from the bone marrow of 4–8-week-old C57BL/6 mice that were cared for and maintained in pathogen-free conditions at the Sede de Investigación Universitaria (SIU), de la Universidad de Antioquia. The care and use of the animals were carried out in strict accordance with the recommendations approved by the ethics committee of the SIU.

The bone marrow was obtained as previously described [14]. Cell isolation and purification were performed using a commercial kit [(Cellular Lineage Cell Depletion Kit mouse (Miltenyi Biotec)]. For negative selection, the antibody cocktail was used, namely anti-CD45, anti-CD45R, anti-CD11b, anti-CD5, anti-Gr1 (Ly-6/C), and anti-Ter-119. Subsequently, a positive selection was made using the monoclonal anti-Sca-1, and the purity of the cells was evaluated with a specific antibody for CD105 labeled with APC (allophycocyanin), following the manufacturer’s instructions [12]. The percentage of purity was always above than 78%. Cell viability was determined by trypan blue exclusion (all samples exceeded 80% viability).

HSPCs were transferred to 96-well cell culture plates containing serum-free SFM-34 culture medium (GIBCO) supplemented with 2 mM L-glutamine, 1% penicillin-streptomycin (GIBCO), growth factor of stem cells (SCF, 20 ng/mL), and Flt-3 ligand (FL, 100 ng/mL), and incubated at 37 °C with 5% CO_2_ [12].

### 2.3. Determination of the TLR2, TLR4, and Dectin-1 Expression on HSPC

To determine the expression of the TLRs and Dectin-1, flow cytometry was performed. The cells obtained via the cell purification procedure were stained with specific antibodies for TLR2 (anti-TLR2 conjugated with alexa fluor 647), TLR4 (anti-TLR4-Phycoerythrin—PE), and Dectin-1 (anti-dectin-1 conjugated with PE).

### 2.4. Histoplasma Capsulatum Yeasts

In this study, the *H. capsulatum* 1980 strain (Hc1980) which was obtained from a blood culture from a Colombian patient with HIV and disseminated histoplasmosis, was used. This fungal strain was cultivated under the conditions described elsewhere [15]. Yeast cells were inactivated by using an aliquot of 20 × 10^6^ yeast/mL and re-suspended in a CytoFix fixation buffer (BD, Pharmingen, San Diego, CA, USA) containing 4% paraformaldehyde, for one hour. Then, fungal cells were washed with PBS and adjusted to the desired number and volume for subsequent experiments [6,12,13]. Live *H. capsulatum* yeast cells were only used for the microbicidal assays.

### 2.5. Culture of HSPC with H. capsulatum Yeasts

HSPCs were cultured with inactivated *H. capsulatum* yeasts at a multiple of infection (MOI) of 5 (5 yeasts: 1 HSPC) and incubated for 24 h or 48 h. Additionally, some of the cultures were previously treated with blocking antibodies specific for TLR2 [anti-mouse/human antibody CD282, (Biolegend, Cat. 121801, San Diego, CA, USA)] and TLR4 [complex anti-mouse antibody CD284/MD2, (Biolegend, Cat. 117607)], or with a specific peptide (CLEC7A) for Dectin-1 (Sigma, Aldrich, Cat. SBP3500794, Burlington, MA, USA), all with the aim of determining the participation of these receptors in the recognition of *H. capsulatum* yeasts. Three independent experiments were performed, each in triplicate.

### 2.6. Phagocytosis Assay

To determine the phagocytic capacity of HSPCs, *H. capsulatum* yeasts were inactivated and stained with fluorescein isothiocyanate (FITC), as previously described [15]. HSPCs were stimulated with *H. capsulatum* yeasts previously stained with FITC for an incubation period of one to three hours, and each of the samples was analyzed by flow cytometry. To differentiate yeasts adhered to the surface of stem cells from internalized ones, 0.2% trypan blue was added to quench extracellular fluorescence. The phagocytosis percentage was determined according to the number of FITC positive cells or by the increase in fluorescence intensity using the modified protocol of Megías et al. [6].

### 2.7. Microbicidal Activity of HSPCs against H. capsulatum Yeasts

HSPCs were infected with *H. capsulatum* yeasts, and, after 24 h of co-culture, the microbicidal activity was evaluated. To do this, 800 µL of cold sterile distilled water was added to the wells to lyse the cells (final volume of 1000 μL), and 500 µL of this suspension was used to count colony-forming units (CFU) in BHI solid culture medium with 1% of glucose in three dilutions 1:10, 1:100, and 1:1000 (modified from Rodríguez et al.) [14].

### 2.8. Determination of the Expression of Cytokines and Inflammatory Mediators by Real-Time Quantitative PCR (qPCR)

To determine activation through the expression of genes encoding inflammatory mediators by HSPC, RNA was obtained from cell cultures stimulated with *H. capsulatum*. The processing and treatment of the samples were carried out using the commercial kit E.Z.N.A. [Total RNA Kit I (Omega Bio Tek)] and following the manufacturer’s instructions. The changes in the expression of the mRNA of the target gene were quantified in relation to the expression of the gene that codes for glyceraldehyde-3-phosphate-dehydrogenase (GAPDH). The relative expression was calculated using the 2^−∆∆Ct^ method. Specific primers were used to determine the relative expression of genes encoding the cytokines IL-1β, IL-6, IL10, IL-17, TNF-α, and TGFβ, as well as arginase-1 and inducible nitric oxide synthase (iNOS) (Table 1).

### 2.9. Apoptosis Assay

The percentages of apoptosis and necrosis were determined by flow cytometry from the cultures of the HSPC with *H. capsulatum* yeasts. Samples were labeled with FITC-coupled Annexin V and propidium iodide using the commercial ApoDetect kit (Invitrogen Corporation, Carlsbad, CA, USA) according to the manufacturer’s instructions.

### 2.10. Cell Proliferation Assay

To assess HSPC proliferation in response to different stimuli, a commercial 5-bromo-2-deoxyuridine [(BrdU) (BD kit, Pharmingene, San Diego, CA, USA)] was used. The incorporation capacity of BrdU in HSPC was determined after being stimulated with *H. capsulatum* yeasts and specific ligands for TLR2 (Pam3Cys), TLR4 (Lipopolysaccharide—LPS), and Dectin-1 (Betaglucan); subsequently, the cells were fixed and evaluated by flow cytometry following the manufacturer’s instructions.

### 2.11. Statistical Analysis

Data analysis was performed with Graph Pad Prism version 7 software (GraphPad Software, Inc., La Jolla, CA, USA). For all values, normality was verified by the Shapiro-Wilk test. Means and standard deviations (SD) were used to determine the percentage of phagocytosis, and medians for flow cytometric analyzes and cytokine expression levels. The Mann-Whitney test was used for comparisons between groups in all the methodologies developed. *p* values < 0.05 were considered significant.

## 3. Results

### 3.1. Identification of the Bone Marrow-Derived HSPC Population

The identification of the HSPC was carried out from the cell purification procedure by columns and specific antibodies coupled to magnetic beads in two steps. After positive selection, cells were labeled with anti-CD105 and Sca-1 and their expression was determined by flow cytometry. A population called Long Term Culture Initiating-Cells (LTC-IC) was obtained, which is characterized as CD105^+^ Sca-1^+^ (Figure 1).

### 3.2. Histoplasma Capsulatum induces Increased Expression of TLR2 and Dectin-1 in HSPC

The expression of membrane receptors in HSPCs was evaluated by flow cytometry; the level of expression of receptors is shown as mean fluorescence intensity (MFI). HSPCs were shown to have a high basal expression of TLR2, TLR4, and Dectin-1; unstained CD105^+^ Sca-1^+^ cells were used as the negative control. Subsequently, it was determined whether *H. capsulatum* yeasts induced the expression of these receptors, and it was observed that the fungus induces a significant increase in the number of receptors for TLR2 and Dectin-1, but not for TLR4 (Figure 2).

### 3.3. HSPCs Are Able to Phagocytose H. capsulatum Yeasts but Do Not Exert a Fungicidal Effect

It was observed that HSPCs have the capacity to adhere to and internalize the yeasts of *H. capsulatum*. These cells showed a percentage of phagocytosis of 17.7% (Figure 3A). Results are shown as a percentage of FITC positive cells. Treatment of HSPC with antibodies specific for TLR2 and TLR4, and the peptide CLEC7A specific for Dectin-1, individually or in combination, was shown to result in a significant decrease in the percentage of phagocytosis in all cases (*p* < 0.0001) (Figure 3B–H). The results suggest that this process is dependent on TLR2, TLR4, and Dectin-1. On the other hand, when evaluating the microbicidal effect of HSPC against the yeasts of *H. capsulatum,* it was observed that the growth of the fungus was not affected (Figure 4).

### 3.4. Histoplasma Capsulatum induces the Expression of Cytokines and Inflammatory Mediators in HSPCs

The expression of cytokines associated with the inflammatory response in HSPC cultures stimulated with *H. capsulatum* yeasts was evaluated by qPCR, and it was observed that the fungus significantly induces the expression of genes that code for IL-1β, IL-6, IL10, IL-17, TNF-α, and TGF-β, as well as the genes that code for the enzymes Arg-1 and iNOS (*p* < 0.0001) compared to the expression of such mediators in uninfected cells (Figure 5). Interestingly, the receptor blockade significantly decreased the expression of these mediators (*p* < 0.0001), especially when the specific antibody against TLR4 and the peptide CLEC7A (specific for Dectin-1) were used, alone or in combination. These results suggest that the expression of these inflammatory mediators is dependent on TLR2, TLR4, and Dectin-1.

### 3.5. H. capsulatum Yeasts Induce Apoptosis and Necrosis and Affect Proliferation in HSPCs

The apoptosis assay showed that *H. capsulatum* yeasts are able to induce apoptosis in 25.4% and necrosis in 2.60% in HSPCs (Figure 6). Additionally, it was observed that the fungal yeasts alter the proliferation of HSPCs depending on the receptor with which they interact. Cells stimulated with the specific ligands for TLR2 (Pam3), TLR4 (LPS), and Dectin-1 (β-glucan) were used as a positive control, and cells cultured without stimuli were used as a negative control. Pam3, LPS, and β-glucan ligands were found to induce HSPC proliferation by 16.1%, 17.5%, and 25.4%, respectively (Figure 7A–D). Similarly, it was shown that the cells simultaneously stimulated with the specific ligands of TLR4 and the yeasts of *H. capsulatum* have a significant increase in the percentage of the population that is in phase G2 or mitosis (G2-M) of a 9.9% (*p* < 0.0001) (Figure 7F). In contrast, the stimulation of β-glucan and fungal yeasts induced a decrease in proliferation by 8.6% (*p* < 0.01) (Figure 7G). In the case of stimulation with the ligand for TLR2, the fungal yeasts did not affect the proliferation process (Figure 7E).

## 4. Discussion

Hematopoiesis is the process by which HSCs give rise to mature cells. This process is regulated by extrinsic and intrinsic factors (cytokines, soluble factors, chemokines, and cell-cell interactions) that determine the loss of differentiation potential and the gain of lineage-specific functions [16]. After an infectious process, there may be an expansion of HSPC by demand, or, on the contrary, a decrease in these cells may occur due to the depletion of reserves due to mechanisms such as apoptosis or necrosis [8,17].

The initial recognition of the pathogen by the cells of the host occurs by the detection of specific molecules of the microorganism; this group of molecules is known as pathogen-associated molecular patterns (PAMPs), which, in turn, are recognized by their corresponding receptor [4]. In this sense, the present study determined the effect of *Histoplasma capsulatum* yeasts recognition by HSPCs, and it was observed that these cells express TLR2, TLR4, and Dectin-1 receptors under basal conditions and that after stimulation with fungal yeasts, a significant increase in the number of TLR2 and Dectin-1 receptors is evident. These results coincide with previous findings described by Megías et al. (2016), who reported that HSCs are targets of fungal pathogens such as *Candida albicans* and that TLR2 and Dectin-1 play an important role in hematopoiesis in response to infection [6].

Similarly, it has been described that TLR2 and Dectin-1 recognize the Ysp3 protein and the 1-3-β glucan present in the wall of *H. capsulatum,* and that they are involved in the defense of the host against infection [18,19]. In this study, it was shown that HSPCs are able to phagocytose *H. capsulatum* yeasts; however, this process does not affect the survival of the fungus. Some similar studies have shown the phagocytic capacity of HSPCs against pathogens such as *Leishmania infantum* and *Candida albicans* [20,21]. During *Histoplasma* infection, yeasts can proliferate intracellularly, which is why they have been described as promoting phagocytosis through the receptor of complement [22]. However, other studies have shown in-vitro that TLR2 and Dectin-1 are involved in the phagocytosis of *H. capsulatum* and *Aspergillus fumigatus* by macrophages [22,23,24]. It has also been established that the conidia of *Scedosporium apiospermum* are recognized by TLR4 and that as a result of this interaction, the macrophages activate with an increase in the production of IL-10 [25]. The results of the present study showed that the blockade of the TLR2, TLR4, and Dectin-1 receptors significantly decreases the percentage of phagocytosis, which seems to indicate that this process is, in part, dependent on these three receptors.

Signaling through *Toll*-like and C-lectin-like receptors is common in response to interaction with fungi. Some studies show an increase in in-vitro production of TNF-α and IL-6 in macrophages activated by *H. capsulatum* through Dectin-1 [26]. Consistent with our findings, it has been observed that the initial immune response against *H. capsulatum* is associated with the production of proinflammatory cytokines such as TNF-α, GM-CSF, IL-23, IL-17, and IL-1β [27]. On the other hand, it has also been established that the production of IL-6, TGF-β, and IL-10 by macrophages is associated with a poor response to infection [28]. The present study found that the increase in the expression of genes that code for IL-1β, IL-6, IL10, IL-17, TNF-α, and TGF-β could be related to the initial recognition of the pathogen and signaling mediated by TLR2 and Dectin-1.

An increase in the expression of the gene that codes for the enzyme iNOS was also observed, which generally acts as a fungistatic against yeasts; however, it has been described that *H. capsulatum* has effective virulence factors that are effective against the generation of nitric oxide, such as the production of nitric oxide reductases, which regulate the conversion of nitric oxide to nitrous oxide, avoiding its toxic effect [29]. Additionally, it was observed that *H. capsulatum* induces the expression of the enzyme Arginase-1 (Arg-1), which competes for L-arginine, also a substrate of iNOS. Arg-1 converts L-arginine to L-ornithine, which is a polyamine used for the cell proliferation of pathogenic dimorphic fungi such as *Coccidioides* spp. [30,31]. In this sense, although the fungus induces the expression of iNOS, it also induces the expression of Arg-1: thus, two antagonistic enzymes could be directly related to the findings. However, a fungicidal effect of HSPCs against *H. capsulatum* was not observed.

Finally, it was confirmed that *H. capsulatum* yeasts induce apoptosis and necrosis in HSPC. Additionally, it was shown that in the presence of specific ligands for TLR2, TLR4, and Detin-1, these cells increase the percentage of cells that enter the G2 or mitosis phase, and that by simultaneously stimulating these cells with the specific ligands and the yeasts of *H. capsulatum*, an increase in the percentage of proliferation is observed only in the presence of LPS. Conversely, in the cells stimulated with β-glucan, the yeasts of *H. capsulatum* decrease the proliferation of HSPC, an effect that could be associated with the increase in the percentage of apoptosis observed in these HSPC.

Previously, Martinez et al. established the role of TLR2 and TLR4 in HSPC signaling in the presence of their soluble ligands and *C. albicans* and demonstrated that TLRs directly influence hematopoiesis, since this interaction induces the proliferation in vitro of HSPC, as well as the differentiation to Lin^+^ cells [32]. Similarly, it has also been established that signaling by Dectin-1 in the presence of β-glucan and *C. albicans* affects hematopoiesis by specifically inducing the differentiation of macrophages with a phenotype altered, tolerated, or trained depending on the stimulus [33].

## 5. Conclusions

In conclusion, these results suggest that the interaction of HSPCs with *H. capsulatum* could induce both changes in the expression of PRRs, specifically TLR2 and Dectin-1, and activate an pro-inflammatory cascade, which, in turn, could affect the viability of these HSPC. Taken together, these findings indicate that *H. capsulatum* could exert a deleterious effect on the hematopoiesis process, which would be reflected in an increase or decrease in leukocytes, erythrocytes, and platelets in patients with severe disease, especially those with dissemination to the bone marrow. Finally, it is important to understand the characteristics, functionality, and factors that regulate the development and differentiation of these cells and the mechanisms that mediate the interaction with fungal pathogens to generate the bases for the improvement and conditioning of cell therapy based on the use of adult stem cells for the treatment of diseases associated with pathogens.

## Figures and Tables

**Figure 1 jof-08-01108-f001:**
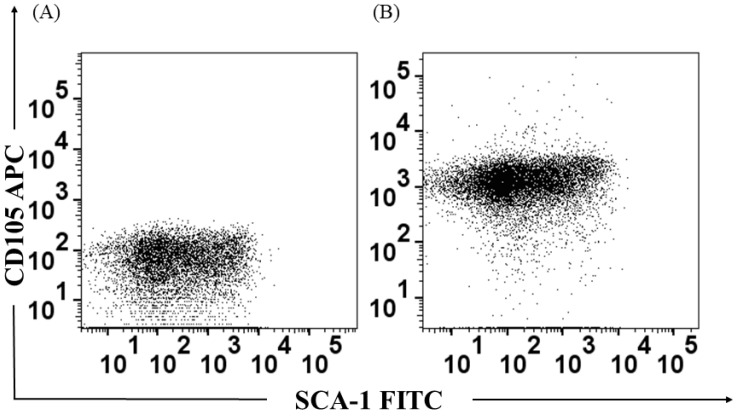
Characterization of the HSPC population. Expression of CD105^+^ and Sca-1^+^ surface antigens on mouse bone marrow long-term culture initiator cells (LT-CIC or LT-HSC). (**A**) Control of Lin^−^ cells without positive selection; (**B**) CD105^+^ Sca-1^+^ cells corresponding to HSPC.

**Figure 2 jof-08-01108-f002:**
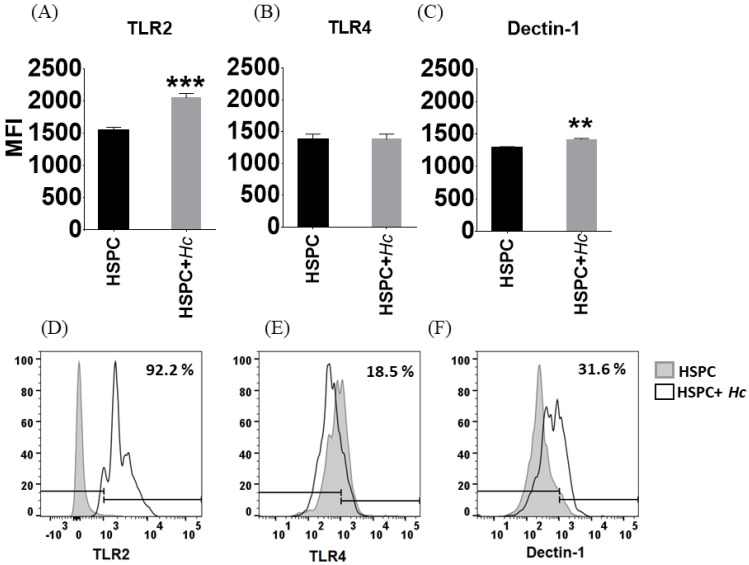
Expression of TLR2, TLR4 and Dectin-1 receptors in HSPC. (**A**,**D**) TLR2 expression in unstimulated and stimulated HSPC with *H. capsulatum* yeasts; (**B**,**E**) TLR4 expression in unstimulated and stimulated HSPC with *H. capsulatum* yeasts; (**C**,**F**) Expression of Dectin-1 in unstimulated and stimulated HSPC with *H. capsulatum* yeasts. HSPCs were selected from the CD105^+^/Sca-1^+^ population and the level of expression of receptors is expressed as mean fluorescence intensity (MFI). (**A**–**D**) Results are expressed as means ± SD of pooled data from three independent experiments. (**D**–**F**) Data represent the percentage of HSPC positive cells and are from a representative experiment of three replicates, ** *p* < 0.01; *** *p* < 0.0001.

**Figure 3 jof-08-01108-f003:**
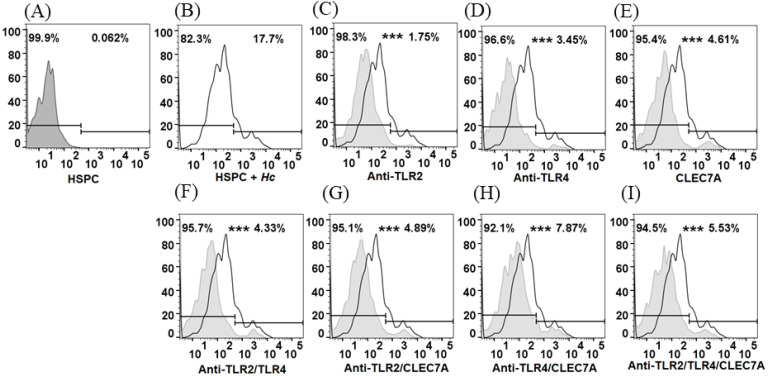
Phagocytosis of *H. capsulatum* yeasts by HSPC. Phagocytosis was analyzed by flow cytometry and the result is expressed as the percentage of FITC positive cells (% phagocytosis) of HSPC cells infected with *H. capsulatum* yeasts. (**A**) Control, HSPC (**B**) Control, percentage of phagocytosis in HSPC co-cultured with *H. capsulatum* and without treatment; (**C**) percentage of phagocytosis in HSPC treated with anti-TLR2; (**D**) percentage of phagocytosis in HSPC treated with anti-TLR4; (**E**) percentage of phagocytosis in HSPC treated with the peptide CLEC7A; (**F**) percentage of phagocytosis in HSPC treated with the anti-TLR2/anti-TLR4 combination; (**G**) percentage of phagocytosis in HSPC treated with the anti-TLR2/CLEC7A combination; (**H**) percentage of phagocytosis in HSPC treated with the combination of anti-TLR4/CLEC7A, and (**I**) percentage of phagocytosis in HSPC treated with the combination of anti-TLR2/anti-TLR4/CLEC7A. Histograms in gray correspond to HSPC previously treated with blocking antibodies for TLR or with a specific peptide for Dectin-1 (CLEC7A). Data are representative from an experiment of three replicates; *** *p* < 0.0001.

**Figure 4 jof-08-01108-f004:**
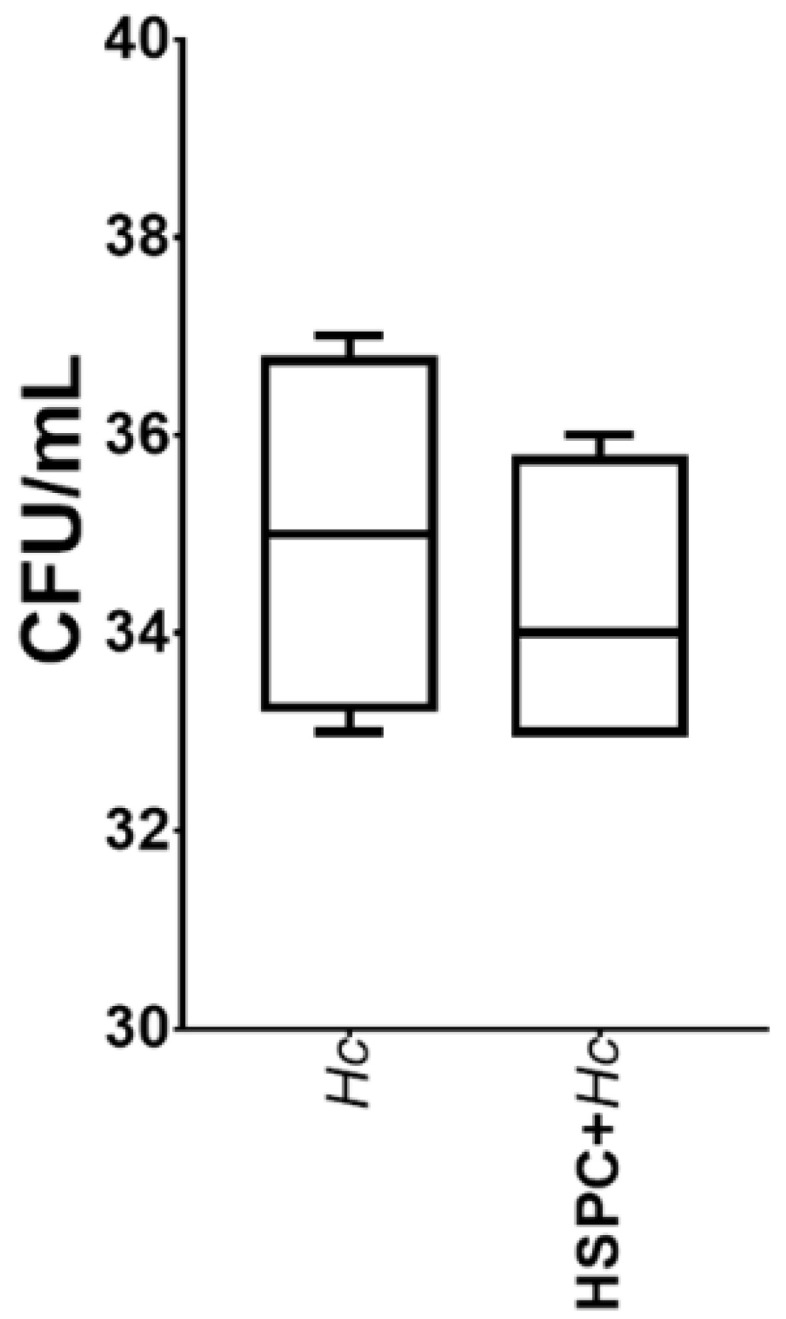
Fungicidal activity of HSPC against *H. capsulatum*. Colony forming units (CFUs) were recovered from HPSC infected with *H. capsulatum* yeasts after incubation for 24 h at 37 °C. Results are expressed as median and IQR of pooled data from three independent experiments.

**Figure 5 jof-08-01108-f005:**
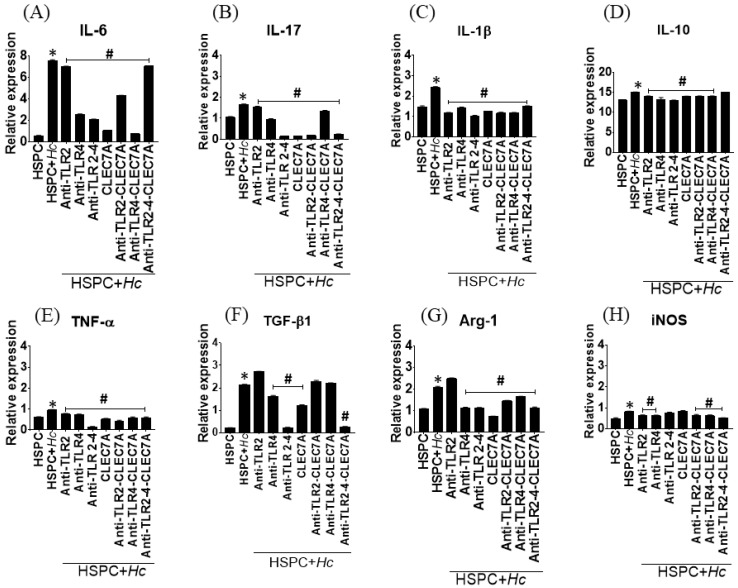
Expression of cytokines and inflammatory mediators in HSPC stimulated with *H. capsulatum* yeast. Analysis of mRNA expression of proinflammatory cytokines, arginase-1, and iNOS in HSPC stimulated or not stimulated with *H. capsulatum* yeast. (**A**) IL-6; (**B**) IL-17; (**C**) IL-1β; (**D**) IL-10; (**E**) TNF-α; (**F**) TGF-β1; (**G**) Arg-1; and (**H**) iNOS. HSPC, control, unstimulated cells; HSPC + Hc, cells stimulated with *H. capsulatum*; TLR, *Toll*-like receptor; CLEC7A, peptide blocker specific for Dectin-1. Results are expressed as means ± SD of pooled data from three independent experiments; * *p* < 0.0001, comparisons were done between HSPCs + *Hc* vs. HSPCs, and ^#^
*p* < 0.0001 HSPCs + *Hc* plus the different treatments vs. HSPCs + *Hc*.

**Figure 6 jof-08-01108-f006:**
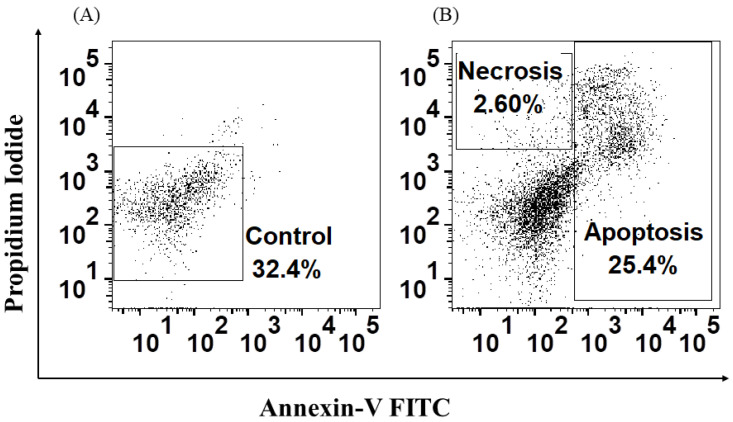
*Histoplasma capsulatum* induces apoptosis and necrosis in HSPC. HSPCs were treated with Annexin V-FITC and propidium iodide as described in materials and methods. (**A**) Control, uninfected HSPC; (**B**) HSPC stimulated with *H. capsulatum* yeasts. Percentages represent the number of cells positive for FITC and propidium iodide. Similar results were obtained from three independent experiments.

**Figure 7 jof-08-01108-f007:**
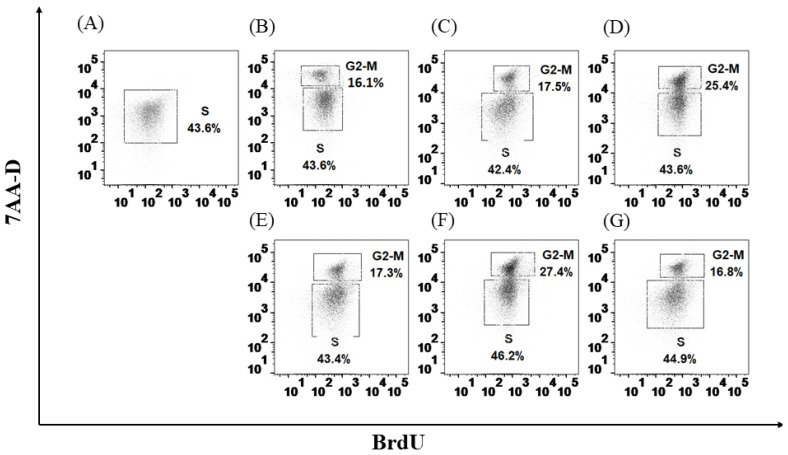
*H. capsulatum* yeasts affect HSPC proliferation. (**A**) Control, unstimulated HSPC; (**B**) HSPC + Pam3CysOH; (**C**) HSPC + LPS; **(D**) HSPC + β-glucan; (**E**) HSPC + Pam3CysOH + *H. capsulatum*; (**F**) HSPC + LPS + *H. capsulatum*; (**G**) HSPC + β-glucan + *H. capsulatum*. Data represent the percentage of BrdU positive cells; results are from a representative experiment of three replicates.

**Table 1 jof-08-01108-t001:** Sequences of the primers or primers for genes that code for cytokines and pro-inflammatory mediators.

Gene	Forward (5′-3′)	Reverse (5′-3′)
GAPDH	CATGGCCTTCCGTGTTCCTA	GCGGCACGTCAGATCCA
IL-6	CAACCACGGCCTTCCCTACTTC	TCTCATTTCCACGATTTCCCAGAG
IL-17	CCAAACACTGAGGCCAAGGACTTC	GGTGACGTGGAACGGTTGAGGTAG
IL-1β	CTTCAAATCTCGCAGCAGCACATC	TCCACGGGAAAGACACAGGTAGC
IL-10	TGGGTTGCCAAGCCTTATCGG	CTCACCCAGGGAATTCAAATGCTC
TNF-α	GACAAGGCTGCCCCGACTACG	CTTGGGGCAGGGGCTCTTGAC
TGFβ	TACTGCCGCTTCTGCTCCCACTCC	TCGATGCGCTTCCGTTTCACCAG
Arg-1	CCTTGGCTTGCTTCGGAACTCA	CTTGGGAGGAGAAGGCGTTTGC
iNOS	GCCGCATGAGCTTGGTGTTTG	GCAGCCGGGAGTAGCCTGTGT

## Data Availability

Not applicable.

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
