# Peer review of "Histoplasma capsulatum Activates Hematopoietic Stem Cells and Their Progenitors through a Mechanism Dependent on TLR2, TLR4, and Dectin-1"

_jof, 2022, doi:10.3390/jof8101108_

Round 1

Reviewer 1 Report

Dear Editor and Authors,

I am glad to review a manuscript with high significance in the research field related to invasive fungal infections. I would require a major revision to clarify some data that should be described with more details and also explain the controls used to support the findings. So far, I am not comfortable to agree with the conclusion. Please, find attached the manuscript revised that I highlighted in the main text my concerns and also added some comments.

Author Response

Dear Reviewer,

We appreciate the constructive nature of your comments and suggestions aimed at improving our paper.

In the new version presented here, we incorporated all the suggestions made by you. We hope that the new revised version of this manuscript will be considered acceptable for publication in the Journal.

In attached file you will find the rebuttal letter.

Reviewer 2 Report

In the present study, the interaction between HSPCs and H. capsulatum yeasts was investigated using in-vitro assays. Results showed that these cells have the capacity to phagocytize the fungal yeasts, without affecting the survival of the fungus. The interaction of H. capsulatum with HSPCs also induces a significant increase in the expression of TLR2 and Dectin-1 but not of TLR4. Additionally, this interaction increases the expression of IL-1β, IL-6, IL-10, IL-17, TNF-α, and TGF-β, as well as the immune mediators Arg-1 and iNOS. It was also established that the fungal yeasts induced apoptosis  and altered the proliferation of HSPCs.

The study has the scientific merit necessary for its publication. It is well-written, results are really very interesting and are well-presented. The methodology is adequate for the study. 

Author Response

Dear Reviewer,

We appreciate the constructive nature of your comments and suggestions aimed at improving our paper.

In the new version presented here, we incorporated all the suggestions made by you. We hope that the new revised version of this manuscript will be considered acceptable for publication in the Journal.

In the present study, the interaction between HSPCs and H. capsulatum yeasts was investigated using in-vitro assays. Results showed that these cells have the capacity to phagocytize the fungal yeasts, without affecting the survival of the fungus. The interaction of H. capsulatum with HSPCs also induces a significant increase in the expression of TLR2 and Dectin-1 but not of TLR4. Additionally, this interaction increases the expression of IL-1β, IL-6, IL-10, IL-17, TNF-α, and TGF-β, as well as the immune mediators Arg-1 and iNOS. It was also established that the fungal yeasts induced apoptosis and altered the proliferation of HSPCs.

The study has the scientific merit necessary for its publication. It is well-written; results are really very interesting and are well-presented. The methodology is adequate for the study. 

R/ Thank you very much for the nice comments.

Reviewer 3 Report

In the article “Histoplasma capsulatum activates hematopoietic stem cells and two of their progenitors through a mechanism dependent on TLR2, 3 TLR4, and Dectin-1”, the authors determined whether the interaction of HSPCs with H. capsulatum yeast affects hematopoiesis, activation, or proliferation of these cells. The article is interesting and of great application since by understanding the factors that regulate the development and differentiation of HSPC cells, as well as the mechanisms that mediate the interaction with fungal pathogens, the bases for improving and conditioning therapy can be generated. based on the use of adult stem cells for the treatment of diseases associated with pathogens.

The manuscript is very well prepared, edited and planned. I have some comments:

The authors do not mention the characteristics of the Hc1980 strain, that is, whether it is of high virulence or low virulence since it could be related to the interaction between HSPC and H. capsulatum and, depending on the virulence, could induce changes in the expression of the pattern recognition receptors (PRR).

On the other hand, the results of this work are interesting since they determined the effect of the recognition of H. capsulatum yeasts by HSPC and observed that these cells express the TLR2, TLR4, and Dectin-1 receptors in basal conditions and that after stimulation with fungal yeasts inactivated, there is a significant increase in the number of TLR2 and Dectin-1 receptors. They also showed that HSPCs can phagocytize live H. capsulatum yeasts and that this process does not affect the survival of the fungus. Likewise, they evidenced the expression of genes encoding inflammatory mediators by HSPC after the activation of cells by H. capsulatum. However, I suggest broadening the discussion and justifying, what is the relevance of using inactivated yeasts and live yeasts for each test. Would the results be modified if the yeasts were used differently?

I suggest carefully reviewing the references to homogenize the format according to the journal's guidelines.

Author Response

Dear Reviewer,

We appreciate the constructive nature of your comments and suggestions aimed at improving our paper.

In the new version presented here, we incorporated all the suggestions made by you. We hope that the new revised version of this manuscript will be considered acceptable for publication in the Journal.

In the article “Histoplasma capsulatum activates hematopoietic stem cells and two of their progenitors through a mechanism dependent on TLR2, 3 TLR4, and Dectin-1”, the authors determined whether the interaction of HSPCs with H. capsulatum yeast affects hematopoiesis, activation, or proliferation of these cells. The article is interesting and of great application since by understanding the factors that regulate the development and differentiation of HSPC cells, as well as the mechanisms that mediate the interaction with fungal pathogens, the bases for improving and conditioning therapy can be generated, based on the use of adult stem cells for the treatment of diseases associated with pathogens.

The manuscript is very well prepared, edited and planned. I have some comments:

The authors do not mention the characteristics of the Hc1980 strain, that is, whether it is of high virulence or low virulence since it could be related to the interaction between HSPC and H. capsulatum and, depending on the virulence, could induce changes in the expression of the pattern recognition receptors (PRR).

R/ The Histoplasma capsulatum 1980 strain was isolated from a Colombian patient and is part of the Corporación para Investigaciones Biológicas (CIB, a Colombian research institution) collection. Although its genome has already been sequenced, it has not yet been published, so it is not possible to reference it. However, this strain has been used in other studies with animal models [López LF, Muñoz CO, Cáceres DH, Tobón ÁM, Loparev V, Clay O, Chiller T, Litvintseva A, Gade L, González Á, Gómez BL. Standardization and validation of real time PCR assays for the diagnosis of histoplasmosis using three molecular targets in an animal model. PLoS One. 2017;12(12):e0190311. doi: 10.1371/journal.pone.0190311].

Information about its precedence has been included in the materials and methods section.

Regarding its virulence, this strain has been considered as intermediate virulence; however, with this scarce information, we cannot hypothesize if this affects the interaction with PRRs or induces changes in their expression.

On the other hand, the results of this work are interesting since they determined the effect of the recognition of H. capsulatum yeasts by HSPC and observed that these cells express the TLR2, TLR4, and Dectin-1 receptors in basal conditions and that after stimulation with fungal yeasts inactivated, there is a significant increase in the number of TLR2 and Dectin-1 receptors. They also showed that HSPCs can phagocytize live H. capsulatum yeasts and that this process does not affect the survival of the fungus. Likewise, they evidenced the expression of genes encoding inflammatory mediators by HSPC after the activation of cells by H. capsulatum. However, I suggest broadening the discussion and justifying, what is the relevance of using inactivated yeasts and live yeasts for each test. Would the results be modified if the yeasts were used differently?

R/ Thank you very much for the comment.

It is important to note that only live yeasts were used in the microbicidal assays for obvious reasons. In the other tests, the yeasts were used inactivated for several reasons: i) Histoplasma is considered a class III biosafety pathogen, so it cannot be used in the flow cytometry core of the Universidad de Antioquia for biosafety reasons; ii) the use of inactivated microorganisms are widely used in interaction studies, since their growth could mask the effects of such interactions; and iii) the use of live microorganisms, in this case yeasts, could in one way or another affect the interaction with the host cells, for example with the production and secretion of soluble molecules that could have an effect beyond the interaction with cellular receptors, for example, block phagocytosis, protein or receptors degradation or affect other activation mechanisms.

I suggest carefully reviewing the references to homogenize the format according to the journal's guidelines.

R/ Thank you for the comment, the references were formatted according to the journal's guidelines.

Round 2

Reviewer 1 Report

Dear authors and co-authors,

Please, find attached the file revise with my comments highlighted in yellow.

Best

Author Response

Reviewer No. 1

Please, rephrase the text highlighted using short sentences.

R/ Thanks for the suggestion

The text was rephrased as suggested

The authors must indicate the tissue and more details about this clinical isolate.

R/Thanks for the comment.

More details were added about this clinical isolate

Please, the authors could add histograms and also a overlap between HSPC and HSPC+Hc for each condition. This comment was considered in the Round 1 and the authors did not answer.

R/ Thanks for the suggestion.

The requested histograms were added.

Please, the authors should add the histograms of HSPC alone to demonstrate the shift when H capsulatumwas considered.

R/ Thanks for the suggestion.

The histogram of HSPC alone was added.

Please, check the subtitle in x-axys.

R/ Thanks for the comment.

The subtitle in both the x- and y-axes was corrected.

The authors could discuss why the groups HSPC+Hc+treatment were not compared to HSPC+treatment, because the effect of treatment alone in the relative expression can be missed.

R/ These experiments to determine gene expression in untreated cells alone were not performed for several reasons:

  1. The antibodies used are blocking monoclonal antibodies, and it is known that this type of antibodies does not have an effect when binding with its respective ligand.
  2. On the other hand, the expression of the genes, observed in the untreated cells alone, was minimal, indicating that the use of blocking antibodies would have no effect.
  3. Additionally, as observed in the experiments using the stimulated cells with fungal yeast, the blockade of the receptors prior to the stimulation with the fungus, decreases the expression of the inflammatory mediators genes induced by Histoplasma, which clearly indicates that the interaction of the fungus with these receptors, it is the mechanism that induces the expression of the inflammatory mediators. With all due respect, we believe that these results (as a control) would not provide additional results.

Please, check if any information was missed.

R/ Thanks for the observation

It was corrected

This conclusion should be supported by control group (HSPC+treatment) showing that the treatment did not reduce the relative expression even in the absence of fungi.

R/ Thanks for the observation.

Please see the comment before.
